# Electric Properties of Molecule Zr_2_Fe Based on the Full Relativistic Theory

**DOI:** 10.3390/molecules24061127

**Published:** 2019-03-21

**Authors:** Jiangfeng Song, Li Zhang, Xianggang Kong, Xiaoyu Hu, Daqiao Meng, Zhenghe Zhu

**Affiliations:** 1Institute of Atomic and Molecular Physics, Sichuan University, Chengdu 610065, China; iterchina@163.com (J.S.); lizhang@scu.edu.cn (L.Z.); mengdaqiao@caep.cn (D.M.); 2China Academy of Engineering Physics, P.O. Box 9072, Jiangyou 621908, China; 3College of Optoelectronic Technology, Chengdu University of Information Technology, Chengdu 610225, China; kxg@cuit.edu.cn; 4University of South China, Hengyang 421001, China; 13990105175@163.com

**Keywords:** Dirac equation, electric dipole moment, electric field gradients, Zr_2_Fe

## Abstract

The present work is devoted to the study of the electric properties: electric dipole moment, electric quadrupole moment, electric field gradients and electric dipole polarizability of molecule Zr_2_Fe on base of the full relativistic theory with basis set 3–21G. The electric dipole moment of Zr_2_Fe is symmetrical to the axis of $C_{2V}$—the vector sum of two projections for two chemical bond FeZr (3.2883 Å), based on when there is charge distribution. The force constant $k_{2}$ is directly connected with electric field gradients.

## 1. Introduction

Zr_2_Fe was supposed to be an important choice for hydrogen storage materials due to its large hydrogen absorption capacity and extremely low hydrogen pressure equilibrium at room temperature and has important applications in the field of tritium recovery in fusion reactors [1,2,3]. Due to its harsh working environment in the fusion reactor, it is necessary to conduct a detailed study of its various aspects [4,5,6]. At present, a good deal of research on its hydrogen absorption performance has been carried out experimentally, but the nature of electricity has not been reported. Among them, electrical properties are one of the important properties of molecules, and some of its early experimental and theoretical work, especially the pioneering research of P. Debye [7,8] is very meaningful. From electrical properties such as dipole moment, quadrupole moment, electric field gradient and polarizability, the symmetry of the molecule, the volume and the electrostatic force within the molecule can be determined. There are two types of methods for expanding the multipole moments of a molecule, namely the plane wave method and the irreducible tensor operators (ITO). The interacting boson approximation model (IBM) play an important role [9,10]. The ITO method can not only directly derive the selection rules of electronic state transitions, but also easily calculates directly using the Wigner–Eckart theorem. The group theory method can only give the selection rule of the electronic state transition, and the ITO method has more advantages. The multipole moment spread of the charge distribution by the ITO method is [11]
(1)$$Q_{q}^{k}=e[\frac{4\pi }{\left(2k+1\right)}{]}^{1/2}r^{k}Y_{q}^{k}(\theta_{e},\phi_{e})=er^{k}C_{q}^{k}(\theta_{e},\phi_{e})$$
$C_{q}^{k}(\theta ,\phi )$ is Racah’s normalized spherical harmonic operator, which is related to the spherical harmonic function.
(2)$$C_{q}^{k}(\theta ,\phi )=[\frac{4\pi }{\left(2k+1\right)}{]}^{1/2}Y_{q}^{k}(\theta ,\phi )$$

$Q_{q}^{k}$ is tensor operator. When *k* = 0, 1, 2, 3, ...., it is called electric monopole moment, electric dipole moment, electric quadrupole moment, electric eight pole moment, etc.

When *k* = 1, it is the electric dipole moment, and its *q* = 1, 0, −1, which are respectively
(3)$$Y_{1}^{1}=-(\frac{3}{8\pi }{)}^{1/2}\frac{\chi +iy}{r};\;Y_{0}^{1}=(\frac{1}{4\pi }{)}^{1/2}\frac{z}{r};\;Y_{-1}^{1}=(\frac{3}{8\pi }{)}^{1/2}\frac{\chi -iy}{r}$$

For the diatomic molecule, the molecular axis is the z-axis, and the electric dipole moment vector coincides with the molecular axis. Therefore, only the z component ($Q_{q}^{k}=Q_{0}^{1}=\mu_{0}^{1}=e_{z}$) of the electric dipole moment is not zero.

For such a diatomic molecule, the electric quadrupole moment is the second-order tensor, and its component is
$$Y_{2}^{2}(\theta ,\phi ),\;Y_{1}^{2}(\theta ,\phi ),\;Y_{0}^{2}(\theta ,\phi ),\;Y_{-1}^{2}(\theta ,\phi ),\;\mathrm{and}\;Y_{-2}^{2}(\theta ,\phi )$$

It can be shown that only the three components on the main diagonal, i.e., $Q_{xx}$, $Q_{yy}$, and $Q_{yy}$ are not zero.

For the polarization rate *α*, it is also a second-order tensor. But by the formula
(4)$$\mu =\alpha \vec{E}$$

It is known that the electrode formation rate *α* is the electric dipole moment at the unit electric field intensity $\vec{E}$, and dimensional analysis indicates that the dimension of the electrode formation rate is volume, so the electrode polarization rate *α* is actually a quantity.

The time differential perturbed angular correlation study of electric field gradients and their temperature variation were performed at a ^183^Ta radioactive probe in the temperature range 20 K-1173 K [12] and found that only one strong asymmetric quadrupole interaction was observed, of which coupling constant is equal to 1039(8) MHz with asymmetry parameter η = 0.82. In addition, the measurement results show a linear dependence of electric field gradient in contrast to the T^3/2^. The measurement method of atomic and molecular interferometry has been developed, which can be used to assess the reliability of different results from different calculation methods [13]. The electric-field gradient (EFG) at the nucleus for each nonequivalent site of the crystalline compounds Zr_2_Fe were calculated using both the standard first-principles linear muffin-tin orbital method (LMTO) and their developed real-space scheme by Helena M. Petrilli and Sonia Frota-Pessoa [14]. The nonrelativistic method is capable of calculating the dipole moment, the quadrupole moment, and the polarizability [15]. However, the non-relativity theory fails to consider the relativistic effects of heavy elements, and the effects on the electric field gradient and the frequency versus polarization are not yet calculated. Moreover, in non-relativistic theory, the Hamiltonian does not contain spin, that is, the spin is separated from the symmetry of space. Corresponding to LS coupling, orbital angular momentum and spin angular momentum are two independent motion constants, and their symmetry is expressed by a single group. In traditional relativity, spin symmetry has lost its meaning. For example, in non-relativity, H_2_O is a C_2V_ group, and in the theory of relativity, it is replaced by a double group C’_2V_, and a new element $\bar{Ε}$ is introduced. The symmetry element is doubled, and the non-representative number is increased by one. The representation is called the irreducible representation of the fermion, which can be developed by the Fermi function. The irreducible representation of a single group of C_2V_ is called the boson irreducible representation.

In the four-component theory of relativity, if the time reversal operator $\hat{Κ}$ is introduced, which is the inverse unitary operator, it is impossible to have the product of the two operators by two corresponding products, that is, the irreducible unitary representation of the group is not obtained because the unitary representation operator and the inverse unitary representation operator can’t occur at the same time. However, it can still form a set of matrices, which is called a corepresentation, which can be proved that it may still become irreducible representation [16,17,18,19]. A group containing both symmetry of space and time-reversed symmetry is called a fully symmetric group, and the relativity based on a fully symmetric group is called complete relativity. In this paper, the energy and electrical properties of molecular Zr_2_Fe are studied by the theory of Dirac equation based on fully symmetric group.

## 2. Elemental Theory

Equivalent definition of time reversal operator: We use another type, that is, Kramer’s theorem to define the time reversal operator, such as ${\hat{Κ}}^{2}$ = −1, then there is ${\hat{Κ}}^{2}\phi =-\phi =\bar{\phi }$, that is, after two time reversals, the state is restored, but the wave function is reversed; if ${\hat{Κ}}^{2}$ = +1, there is ${\hat{Κ}}^{2}\phi =\phi$, that is, after two time reversals, the state is restored, but the wave function is not inverted. Thus, Kramer can be used to expand the base group, i.e., {ϕ} and $\left\{\bar{\phi }\right\}$ to expand the operator.

Time reversal operator: a time reversal operator or Kramer operator
$\hat{Κ}$, which is defined as
(5)$$\hat{Κ}\psi (r,t)=\psi^{*}(r,-t)$$
If the Hamiltonian operator is a real function of *r*, then there is
(6)$$\hat{Κ}\hat{H}(r)\psi (r,t)={\hat{H}}^{*}(r)\psi^{*}(r,-t)=\hat{H}(r)\psi^{*}(r,-t)=\hat{H}\hat{Κ}\psi (r,t)$$
That is
(7)$$[\hat{H},\hat{Κ}]=0$$

This means that the state is invariant to time reversal, or
$\hat{Κ}$ and $\hat{H}$
have the same eigenfunction.

It can be proved that the time reversal operator is the inverse operator [20,21], i.e.,
(8)$$\langle \hat{Κ}\psi |\hat{Κ}\phi \rangle ={\langle \psi |\phi \rangle }^{*}=\langle \phi |\psi \rangle$$
The same is also an inverse linear operator.

Kramer’s theorem [22]: It can be proved that, if ${\hat{Κ}}^{2}$ = −1, which corresponds to a semi-positive J value, that is, a fermion, the time reversal produces at least a new double degeneracy; if ${\hat{Κ}}^{2}$ = +1, which corresponds to the integer J value, that is, bosons, the time reversal does not produce new double degeneracy.

For the stationary Dirac equation, the operator is
(9)$${\hat{h}}_{D}\psi =Ε\psi ;{\hat{h}}_{D}=\beta^{\prime }mc^{2}+c(\alpha .\hat{p})+\hat{V}$$
where,
(10)$$\alpha =\left[\begin{array}{cc} 0 & \sigma \\ \sigma & 0 \end{array}\right];\;\beta^{\prime }=\left[\begin{array}{cc} 0 & 0\\ 0 & 2{Ι}_{2} \end{array}\right];\;I_{2}\;=\;2\;\times \;2\;\mathrm{unit}\;\mathrm{matrix}$$
and Pauli spin matrix
(11)$$\sigma_{x}=\left[\begin{array}{cc} 0 & 1\\ 1 & 0 \end{array}\right];\;\sigma_{y}=\left[\begin{array}{cc} 0 & -i\\ i & 0 \end{array}\right];\;\sigma_{z}=\left[\begin{array}{cc} 1 & 0\\ 0 & -1 \end{array}\right]$$
If $\hat{V}$ = 0, it will be the Dirac equation of free electrons.

Here, single electronic operator [21,22,23,24] is
(12)$${\hat{h}}_{D}=\left(\begin{array}{cc} A & Β\\ -{Β}^{*} & A^{*} \end{array}\right)=\left(\begin{array}{cccc} \hat{V} & -ic{\hat{d}}_{z} & 0 & -ic{\hat{d}}_{-}\\ -ic{\hat{d}}_{z} & -2mc^{2}+\hat{V} & -ic{\hat{d}}_{-} & 0\\ 0 & -ic{\hat{d}}_{+} & \hat{V} & ic{\hat{d}}_{z}\\ -ic{\hat{d}}_{+} & 0 & ic{\hat{d}}_{z} & -2mc^{2}+\hat{V} \end{array}\right)$$
where
(13)$${\hat{d}}_{z}=\frac{\partial }{\partial z},\;{\hat{d}}_{\pm }=\frac{\partial }{\partial x}\pm i\frac{\partial }{\partial y}.$$
It is easy to know that
(14)$$A=\left(\begin{array}{cc} \hat{V} & -ic{\hat{d}}_{z}\\ -ic{\hat{d}}_{z} & -2mc^{2}+\hat{V} \end{array}\right)=A^{+}$$
(15)$$Β=\left(\begin{array}{cc} 0 & -ic{\hat{d}}_{-}\\ -ic{\hat{d}}_{-} & 0 \end{array}\right)=-{Β}^{+}=\left(\begin{array}{cc} 0 & -ic{\hat{d}}_{+}\\ -ic{\hat{d}}_{+} & 0 \end{array}\right)$$

That is to say, A is Hermitian, that is, after it is used, the wave function is not inverted; B is anti-Hermitian, that is, after it is applied, the wave function is reversed. At this time, there are two sets of Kramer pair base sets, and the Dirac operator ${\hat{h}}_{D}$ reflects time reversal symmetry.

It is easy to prove that ${\hat{h}}_{D}$ can be expressed as [22,23,24]
(16)$${\hat{h}}_{D}\;=\;{Ι}_{2}⊗\left(\begin{array}{cc} \hat{V} & 0\\ 0 & -2mc^{2}+\hat{V} \end{array}\right)-c\overset{\vee }{i}⊗\left(\begin{array}{cc} 0 & {\hat{d}}_{z}\\ {\hat{d}}_{z} & 0 \end{array}\right)-c\overset{\vee }{j}⊗\left(\begin{array}{cc} 0 & {\hat{d}}_{y}\\ {\hat{d}}_{y} & 0 \end{array}\right)-c\overset{\vee }{k}⊗\left(\begin{array}{cc} 0 & {\hat{d}}_{x}\\ {\hat{d}}_{x} & 0 \end{array}\right)$$
Here, the Dirac operator is represented as a quaternion. Quaternion Algebra [25]: The quaternion algebra was developed by Hamilton and Frobenius, but it was limited to quantum mechanics at a very late time. The quaternion can be expressed as
(17)$$q=\sum_{\lambda =0}^{3}V_{\lambda }e_{\lambda }=V_{0}+V_{1}\overset{\vee }{i}+V_{2}\overset{\vee }{j}+V_{3}\overset{\vee }{k}$$
where,
(18)$$e_{1}=\overset{\vee }{i}↔i\sigma_{z};\;e_{2}=\overset{\vee }{j}↔i\sigma_{y};\;e_{3}=\overset{\vee }{k}↔i\sigma_{x}$$

And $\overset{\vee }{i}$, $\overset{\vee }{j}$, $\overset{\vee }{k}$ are the unit number of the quaternion, i is the virtual unit, $V_{0}$, $V_{1}$, $V_{2}$ and $V_{3}$ are the real number, $\sigma_{z}$, $\sigma_{y}$ and $\sigma_{x}$ is the Pauli spin matrix in Equation (11). The quaternion is a four-dimensional vector space composed of a three-dimensional complex space and a one-dimensional real space. From Equation (12) to (18), the use of quaternion algebras establishes a connection with time-reversed symmetry. A group containing spatial symmetry and time-reversed symmetry is called a fully symmetric group, and its representation is called a corepresentation. Equation (16) is a quaternion representation of the quaternion ${\hat{h}}_{D}$, which is the Dirac operator of the fully symmetric group.

Dirac equation in the form of quaternion algebra

(19)h^Dqψq=Εψq

## 3. Theoretical Calculations and Discussions

In this paper, the electrical properties of the molecular Zr_2_Fe are calculated using the relativistic configuration interaction (DIRRCI) method of the DIRAC10 program, and the basis set is 3–21G. The calculation results are listed in Table 1 and Table 2.

The axis of symmetry
$C_{2V}$
of the dipole moment of the molecule is the z-axis, which is the vector sum of the dipole vectors of the two bonds FeZr on the axis of symmetry, and the length of the vector is = 3.2883 Å. Its dipole charge distribution is
(20)$$q=\frac{\mu }{r}=\pm \frac{6.39267291}{3.2883}=\frac{6.39267291*10^{-18}\left(\mathrm{CGSE}.\mathrm{cm}\right)}{3.2883*10^{-8}\left(\mathrm{cm}\right)*3*10^{9}}=\pm \;0.64843/10^{-19}C$$

The dipole moment is a vector, and the above indicates that the charge is unbalanced along the symmetry axis
$C_{2V}$
of the molecule. The unit is 10^−19^ C and C is Coulomb. The quadrupole moment is a second-order tensor, which represents the two-dimensional (vertical symmetry plane
$\sigma_{V}$
) distribution of the charge. For example, the charge distribution of the quadrupole moment of the FeZr molecule is shown in Figure 1, the unit is 10^−19^ C, and C is Coulomb.
(21)$$q_{z}=\frac{Q_{ZZ}}{r}=+\frac{0.0300738}{3.2883}=+\frac{0.0300738*10^{-18}\left(\mathrm{CGSE}.\mathrm{cm}\right)}{3.2883*10^{-8}\left(\mathrm{cm}\right)*3*10^{9}}=+0.003048566/10^{-19}C\;q_{x(y)}=\frac{Q_{ΧΧ(YY)}}{r}=-\frac{0.015036}{3.2883}=\frac{0.015036*10^{-18}\left(\mathrm{CGSE}.\mathrm{cm}\right)}{3.2883*10^{-8}\left(\mathrm{cm}\right)*3*10^{9}}=-0.00152419/10^{-19}C$$

Regarding the relationship between the electric field gradient and the chemical bond, the electric field gradient of diatomic molecule is directly collinear with the axis where the chemical bond is located. For the $C_{2V}$ Zr_2_Fe molecule, the electric field gradient is not collinear with the chemical bond, and the sum of the projections of the electric field gradient component on the chemical bond Fe-Zr is required. Since Zr_2_Fe is in the YZ plane, it acts on the three components of the Fe nucleus: $Q_{XX}$, $Q_{YY}$, and $Q_{ZZ}$. And the projection of its component $Q_{XX}$ on the chemical bond Fe−Zr is zero, and only the projection of $Q_{XX}$ and $Q_{ZZ}$ on the chemical bond Fe−Zr is not zero, as shown in Figure 2.

It is now necessary to calculate the projections acting on the nucleus *Fe* and on the chemical bonds Fe−Zr as follows:

The nucleus Fe electric field gradient (/a.u.) is $Q_{XX}$ = 0.3247777245, $Q_{YY}$ = 0.3247777245 and $Q_{ZZ}$ = 0.6495554490. The component of projection of the Nucleus Fe electric field gradient on the chemical bond Fe−Zr is zero, and only the projections $Q_{ZZ}$ and $Q_{YY}$ on the z and y axes are zero, and the projections are
zz = 0.649555449cos39.078 = −0.649555449 × 0.7762885 = 5.04242425yy = 0.3247777245cos50.922 = 0.3247777245 × 0.69679214 = 0.226302565(22)
For Fe atom, the sum is 5.04242425 + 0.226302565 = 5.2687.

It is now also necessary to discuss the electric field gradient (/a.u.) of the nuclear Zr as $Q_{XX}$ = −0.3115686073, $Q_{YY}$= −0.3115686073 and = 0.6231372146. $Q_{XX}$ component of the projection of the nucleus Zr electric field gradient on a chemical bond is zero, and only the projections on the z and y axes, $Q_{ZZ}$ and $Q_{YY}$, are zero, which are
zz = 0.6231372146cos39.078 = 0.6231372146 × 0.7762885 = 0.483734259yy = −0.3115686073cos50.922 = −0.3115686073 × 0.63037778 = −0.196405926(23)
For Zr atom, the sum is 0.483734259 − 0.196405926 = 0.287328333.

It can be proved that in the direction of the molecular bond, that is, in the z-axis direction, there is one atomic unit electron charge, that is, 1 a.u. charge. And the electric field gradient is 1 a.u., then the second-order force constant = 1 au electron charge * 1 au electric field gradient, that is
(24)$$\begin{array}{l} k_{2}=1\mathrm{au}\;\mathrm{electron}\;\mathrm{charge}*1\mathrm{au}\;\mathrm{electric}\;\mathrm{field}\;\mathrm{gradient}=-4.8029*10^{-10}(\mathrm{CGSE})*9.71736*10^{21}\frac{V}{m^{2}}\\ =-4.8029*10^{-10}(\mathrm{CGSE})*9.71736*10^{21}*\frac{V}{10^{4}\mathrm{cm}}*\frac{1}{300}(\mathrm{CGSE})\frac{1}{\mathrm{cm}}\\ =1.55717*10^{6}\frac{\mathrm{dyne}}{\mathrm{cm}}=1.55717*10^{6}(g\cdot \sec^{-2})=1.55717*10^{6}\frac{g\cdot {\mathrm{cm}}^{2}\cdot \sec^{-2}}{{\mathrm{cm}}^{2}}\\ =1.55717*10^{6}\frac{g.\mathrm{cm}.\sec^{-2}}{\mathrm{cm}}(\frac{\mathrm{dyne}}{\mathrm{cm}})=1.55717*10^{6}(\frac{\mathrm{erg}}{{\mathrm{cm}}^{2}}) \end{array}$$
Zr_2_Fe is a $C_{2V}$ molecule, the axis of symmetry is the Z axis. Fe atom is the origin point, and the molecule is in the YZ plane. Therefore, each FeZr bond is also in the YZ plane. Take each ZrFe bond of Zr_2_Fe as an example. In the vicinity of its equilibrium point, there is 0.270302 electrons move from the Zr atom to the Fe atom. The second-order force constant for each ZrFe bond axis is
(25)$$k_{2}=[\frac{5.2687}{26+0.270302}+\frac{0.287328333}{40-0.270302}]*1.55517*10^{6}\left({\mathrm{ergcm}}^{-2}\right)=3.2315\;*10^{5}\left(\mathrm{dyne}/\mathrm{cm}\right)\;\begin{array}{l} k_{2}=[0.200557+0.007232]*1.55517*10^{6}\left({\mathrm{ergcm}}^{-2}\right)=3.2315*10^{5}\left(\mathrm{dyne}/\mathrm{cm}\right)\\ k_{2}=3.2315*10^{5}\left(\mathrm{dyne}/\mathrm{cm}\right) \end{array}$$
Its resonant frequency is
(26)$$\omega =\frac{1}{2\pi c}(\frac{k}{\mu }{)}^{1/2}=\frac{1}{2*3.1416*3*10^{10}}(\frac{0.32315*10^{6}*6.02*10^{23}}{34.639}{)}^{1/2}\;\omega =\frac{1}{2\pi c}(\frac{k}{\mu }{)}^{1/2}=\frac{1}{2*3.1416*3*10^{10}}(\frac{0.32315*10^{6}*6.02*10^{22}}{3.4639}{)}^{1/2}\;=\;0.039757\;\times \;104^{\;}=\;397.57\;{\mathrm{cm}}^{-1}$$

## 4. Conclusions

In this paper, the electrical properties of the molecular Zr_2_Fe are calculated using the relativistic configuration interaction (DIRRCI) method of the DIRAC10 program, and the basis set is 3–21G. The dipole moment of the molecule along the symmetry axis $C_{2V}$ is the vector sum of the dipole vectors of the two bonds FeZr on the axis of symmetry, i.e., 3.2883 Å, from which the charge fraction is calculated. Unlike linear molecules, their dipole moments coincide with the molecular axis. The electric field gradient is directly related to the twofold force constant
$k_{2}$
of the chemical bond, that is, the second-order force constant
$k_{2}$
= 1 au electron charge * 1 au electric field gradient.

The approximations of present calculations based on DIRAC10 software can be discussed here. Relativistic quantum chemistry is related to solve Dirac equation, whose wave function has four components. The results can be regarded as strict and reliable when the basis set is large enough and omission and selection is suitable during the solving process. Unlikely, the other approximation methods, i.e., nonrelativistic Schrodinger equation including the relativistic effect and inclusion of the relativistic effect in the selected basis sets, are seriously dependent on the systems, which yield much lower reliability [13,14,15,16,17,18,19,20,21,22,23]. In view of the above points, the more powerful and reliable solutions to Dirac equation with DIRAC10 software is applied in present calculations here.

## Figures and Tables

**Figure 1 molecules-24-01127-f001:**
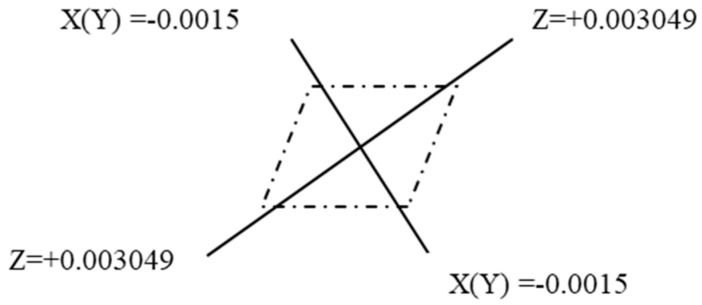
Charge distribution of quadrupole moment of Fe-Zr molecule.

**Figure 2 molecules-24-01127-f002:**
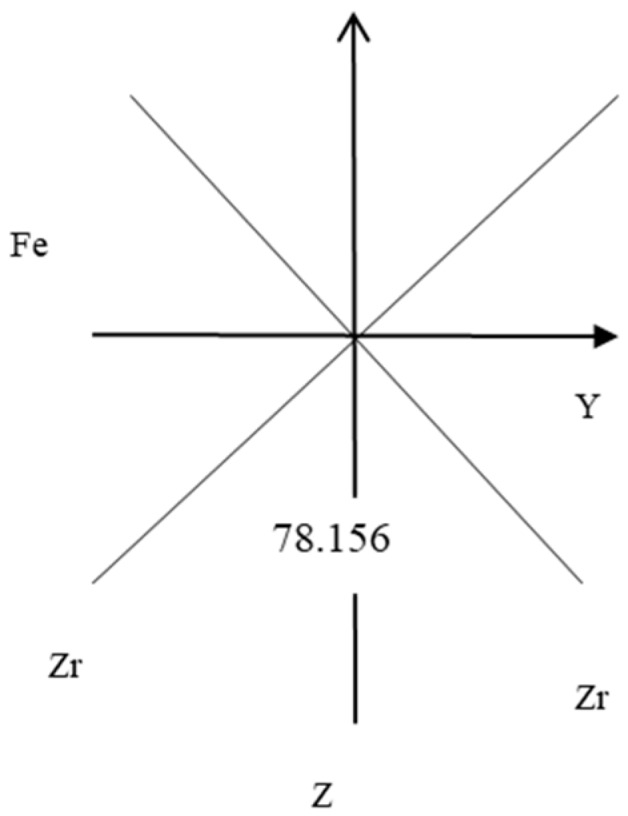
Projection of electric field gradient on chemical bond Fe-Zr.

**Table 1 molecules-24-01127-t001:** Geometry, energy, charge, dipole moment and quadrupole moment of molecular Zr_2_Fe.

Basis Set	3–21G
bond length/Å	2.1188
bond angle/^°^	78.156
Energy/a.u.	−928.75326
energy gap/a.u.	4.03630384
atomic charge	
Fe	−0.540605
Zr	0.270302
Zr	0.270302
dipole moment/Debye	−6.39267291
quadrupole moment/a.u.	$Q_{XX}$ = −0.015036
	$Q_{YY}$ = −0.015036
	$Q_{ZZ}$ = 0.0300738
(dipole moment:a.u. = 2.54177000 Debye)	

**Table 2 molecules-24-01127-t002:** Polarizability and electric field gradient of molecular Zr_2_Fe.

Frequency/a.u.	Polarizability/a.u.	Polarizability/Å^3^	Electric Field Gradient/a.u.
frequency f = 0.00	0.01173850	0.0017395	Nucleus: Zr
frequency f = 0.10	0.01174660	0.001740665	$Q_{XX}$ = −0.311568607
frequency f = 0.20	0.01177109	0.001744294	$Q_{YY}$ = −0.311568607
frequency f = 0.30	0.01181226	0.001750395	$Q_{ZZ}$ = 0.6231372146
frequency f = 0.40	0.01187045	0.0017590176	Nucleus: Fe
frequency f = 0.50	0.01194617	0.0017702382	$Q_{XX}$ = 0.3247777245
frequency:	Polarizability:		$Q_{YY}$ = 0.3247777245
1 a.u. = 2.3497 × 10^8^ HZ	1 a.u. = 0.1482 Å^3^		$Q_{ZZ}$ = 0.6495554490
			(electric field gradient: 1 a.u. = 9.71736 E + 21 V × m^−2^)

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
