# Peer review of "Electric Properties of Molecule Zr2Fe Based on the Full Relativistic Theory"

_molecules, 2019, doi:10.3390/molecules24061127_

Round 1
Reviewer 1 Report
Zr2Fe is one of the main candidate materials for tritium storage system in the International Thermonuclear Fusion Experimental Reactor (ITER) project. The physical properties, such as electrical properties and geometric structure properties of Zr2Fe as a basic molecule of materials, have not been reported in many literatures. These basic properties are very important for material optimization and storage system structure design. This paper chooses Dirac, the international advanced software for relativistic calculation, to study these properties in detail and in-depth with high precision. The electrical dipole moment, electric field gradient and related geometric structure properties of Zr2Fe molecule are given, which shows that the research has considerable scientific significance.The full text method is reliable, the research process is meticulous, the calculation results are credible, and the English writing is fluent. Therefore, we agreed to recommend that the article be published in Molecules.
Author Response
Thank you very much for your reminding.
Reviewer 2 Report
This mansucript describes the theoritical calculation for Zr2Fe-based molecules and the simulated method and results are reasonable and useful. Therefore, I think that this manuscript will be acceptable for several readers in Molecules. However, following contents should be revised.
p.2, line 13. "Y-12(Θ,Φ)和Y-22(Θ,Φ)" is strange.
p.3, The font size of equation (6) is relatively small compared to other equations.
Author Response
Thank you very much for your reminding. We have modified the format.

Reviewer 3 Report
The report of the Manuscript molecules-442527
Title: Electric properties of molecules Zr2Fe based on the full relativistic theory
In this work the Authors represents the results of fully relativistic numerical calculations of some electrical properties of the molecular Zr_2Fe, i.e. the polarizability, the dipole and quadrupole moment and the electric field gradient. Results are given in two Tables and appear to be correct; they may be of interest to the atomic and molecular physics community.
In my opinion, this paper will deserve publication if it is improved in several respects. First of all, the editing of English language is required. Some sentences are grammatically incorrect, which makes the article chaotic and hard to read. For example, on Page 2, in the line 45 one reads: "(...) has only the z component is not zero, that is (...)".
Editing the article as a whole is also not the best. I do not understand, why not all mathematical formulas are numbered. Moreover, the formula in the line 49 contains a Chinese character instead of an English word "and". Why the formula (6) is written in a so small font?
Another thing worth improving is the lack of any details of calculations in Chapter 3. The Authors did not indicate, which formulas has been used to obtain the numbers presented in the Tables 1 and 2. In my opinion, this chapter should be more elaborated, in particular the presentation of the results and their discussion. There are no comparisons to the results obtained by other authors - are they even available in the literature? It should be clearly indicated both in the abstract and Introduction, and of course in the Conclusions.
I recommend publication after a large editing correction.
Author Response
Dear Professor,
Thank you very much for your reply and the comments from the reviewer about our paper submitted to Molecules (molecules-442527)entitled “Electric properties of molecule Zr2Fe based on the full relativistic theory”.
The whole manuscript has been carefully checked and revised according to the comments of the Reviewers, and the relevant revisions were highlighted in red for clearness, which may provide a more readable description on the method and the main results of this study.Now the new version of the revised manuscript as well as a list of the changes is submitted here.
If you have any questions about this manuscript, please don’t hesitate to let us know.
The point-to-point responds to the reviewer’s comments are listed as following.
Yours,
Jiangfeng Song

Reviewer 4 Report
Reviewer's report
Manuscript Title: "Electric properties of molecules Zr2Fe based on the full relativistic theory"
Manuscript ID: molecules-442527
Authors: Song Jiangfeng et al.
The manuscript is devoted to the important problem of calculation of the electric dipole moment, electric quadrupole moment, electric field gradients and electric dipole polarizability. Furthermore, the authors study the molecule Zr2Fe and the presence of the atom Zr requires the use of the relativistic Dirac equation. Therefore, the work is of interest. However, the authors should considerably improve the manuscript to make it suitable for publication in Molecules.
The authors use the original method in order to calculate the electric properties on base of the relativistic Dirac equation. Nevertheless, there are a lot of various methods which allow one to study heavy elements in quantum chemistry (see the books [1,2] and the reviews [3-10]). Many methods are based on reducing the Dirac theory to a two-component form. Specifically, I can mention the methods based on unitary transformations and following the approach elaborated by Douglas, Kroll, and Hess [11,12] but often using different transformation operators [13–17]. These methods allow one to fulfill not only high order [18] but also arbitrary order [14,15,19,20] Douglas-Kroll-Hess transformations. I can also mention the infinite-order two-component method of Barysz and collaborators [21] which is the two-step exact-decoupling approach. Another successful relativistic two-component method is the zeroth-order regular approximation (ZORA) [22]. The exact reduction to the two-component form can also be performed in one step [23]. I believe that the authors should give, at least, a short overview of known relativistic methods and should explain in more details why they follow the approach described in the manuscript. What advantages does it give in relation to the electric properties of molecules containing heavy atoms?
Another shortcoming of the manuscript is poor English. In particular, I can propose to write ``of Molecule Zr2Fe’’ instead of ``of Molecules Zr2Fe’’ in the title.
In the abstract:
Line 13: ``is devoted to the study of the electric properties’’ instead of ``devotes to study the electric properties’’
Line 15: ``on base of the’’ instead of ``based on the’’
Lines 16-17: the phrase ``the vector sum of two projections for two chemical bond FeZr is 3.2883 Å,based on which there are charge distribution.’’ is incomprehensible.
Line 17: ``is directly connected with electric’’ instead of ``is connected with electric’’
Only some examples in the text of the manuscript:
Line 49: Chinese letters should be replaced with English ones
Line 104: ``For the stationary Dirac equation’’ instead of ``For the Dirac equation without time’’
Line 105: ``is the 2X2 unit matrix’’ instead of ``-- 2*2 unit matrix’’
Line 173: ``component’’ instead ``componen’’
Line 231: ``Landau’’ instead of ``Landu’’
All the text should be attentively checked.
In my opinion, the manuscript may be suitable for publication in Molecules after an appropriate revision.
[1] K. G. Dyall and K. Faegri, Introduction to Relativistic Quantum Chemistry (Oxford University Press, Oxford, 2007).
[2] M. Reiher and A. Wolf, Relativistic Quantum Chemistry: The Fundamental Theory of Molecular Science (Wiley-VCH, Weinheim, 2009).
[3] D. Peng and M. Reiher, J. Chem. Phys. 136, 244108 (2012).
[4] D. Peng and M. Reiher, Theor. Chem. Acc. 131, 1081 (2012).
[5] M. Reiher, in Sequential Decoupling of Negative-Energy States in Douglas-Kroll-Hess Theory, Handbook of Relativistic Quantum Chemistry, edited by W. Liu (Springer-Verlag, Berlin, 2015).
[6] J. Autschbach, Coord. Chem. Rev. 251, 1796 (2007).
[7] T. Nakajima and K. Hirao, Chem. Rev. 112, 385 (2012).
[8] M. Reiher, WIREs Comput. Mol. Sci. 2, 139 (2012).
[9] M. Reiher, Theor. Chem. Acc. 116, 241 (2006).
[10] W. Liu, Mol. Phys. 108, 1679 (2010).
Author Response

(The authors gave the same response as above.)

Round 2
Reviewer 4 Report
Reviewer's report 2
Manuscript Title: "Electric properties of molecules Zr2Fe based on the full relativistic theory"
Manuscript ID: molecules-442527
Authors: Song Jiangfeng et al.
The authors have made the required corrections. In my opinion, the revised manuscript is suitable for publication in Molecules.
This manuscript is a resubmission of an earlier submission. The following is a list of the peer review reports and author responses from that submission.